# Does Prior Respiratory Viral Infection Provide Cross-Protection Against Subsequent Respiratory Viral Infections? A Systematic Review and Meta-Analysis

**DOI:** 10.3390/v16060982

**Published:** 2024-06-19

**Authors:** Vennila Gopal, Matthew Chung Yi Koh, Jinghao Nicholas Ngiam, Ong Hang-Cheng, Jyoti Somani, Paul Anatharajah Tambyah, Jeremy Tey

**Affiliations:** 1Yong Loo Lin School of Medicine, National University of Singapore, Singapore 117597, Singaporemdcpat@nus.edu.sg (P.A.T.); 2Faculty of Medicine and Health Sciences, University Malaysia Sabah, Kota Kinabalu 88400, Malaysia; 3Division of Infectious Diseases, Department of Medicine, National University Health System, Singapore 119228, Singapore; matthew.koh@mohh.com.sg (M.C.Y.K.); nicholas.ngiam@mohh.com.sg (J.N.N.); 4Division of Infectious Diseases, Department of Medicine, Faculty of Medicine, University of Malaya, Kuala Lumpur 50603, Malaysia; ong.hc@ummc.edu.my; 5Department of Radiation Oncology, National University Cancer Institute, Singapore 119074, Singapore

**Keywords:** systematic review, meta-analysis, sequential infection, respiratory virus infection

## Abstract

The epidemiology of different respiratory viral infections is believed to be affected by prior viral infections in addition to seasonal effects. This PROSPERO-registered systematic review identified 7388 studies, of which six met our criteria to answer the question specifically. The purpose of this review was to compare the prevalence of sequential viral infections in those with previously documented positive versus negative swabs. The pooled prevalence of sequential viral infections over varying periods from 30–1000 days of follow-up was higher following a negative respiratory viral swab at 0.15 than following a positive swab at 0.08, indicating the potential protective effects of prior respiratory viral infections. However, significant heterogeneity and publication biases were noted. There is some evidence, albeit of low quality, of a possible protective effect of an initial viral infection against subsequent infections by a different virus, which is possibly due to broad, nonspecific innate immunity. Future prospective studies are needed to validate our findings.

## 1. Introduction

Global pandemics of emerging respiratory viruses, such as severe acute respiratory syndrome coronavirus-1 (SARS-CoV-1) in 2003, H1N1 Influenza A in 2009, and the recent SARS-CoV-2 pandemic, have had a major impact on health, social lives, and the economy worldwide. Highly pathogenic coronaviruses, such as the Middle Eastern respiratory syndrome coronavirus (MERS-CoV) or SARS-CoV-1 and 2, in addition to influenza viruses, can cause respiratory failure and, subsequently, death in a high proportion of those infected [1,2]. In addition to pandemics, endemic seasonal influenza and other respiratory viral infections are also associated with high mortality and morbidity rates in elderly patients and those with other respiratory conditions, cardiovascular disorders, or immunosuppressed conditions [3,4,5]. 

SARS-CoV-2, the virus that caused the COVID-19 pandemic, was first detected in Wuhan, China, in December 2019 [6]. This pandemic led to an unprecedented range of disruptions to travel and movement globally, which was associated with a marked reduction in influenza worldwide [7,8,9,10] in 2020–2021. Meanwhile, other respiratory viruses, such as the respiratory syncytial virus (RSV) and rhinoviruses (RV), continued to circulate [11] in many, but not all, countries despite the widespread use of non-pharmaceutical interventions, including masks and quarantines. 

Interestingly, one lineage of influenza B has become practically extinct with the SARS-CoV-2 pandemic; thus, the most recently recommended seasonal influenza vaccine contains only three influenza strains rather than four [12]. The reasons for these observations are unclear. Researchers have noted that individuals with a prior bout of COVID-19 had a lower likelihood of contracting SARS-CoV-2 again [13]. Immunity triggered by infections has been well-recognized for many pathogens, leading to a decrease in epidemic transmission as the pool of susceptible individuals is reduced [14]. Nonetheless, the extent and duration of protective immunity, including cross-immunity to different variants induced by respiratory viral infections, remain inadequately explored. Even less is known about cross-protective immunity against other antigenically distinct viruses.

There are several possible explanations for the epidemiological finding that influenza disappeared in 2020–2021 while other respiratory viruses persisted throughout the pandemic. One is that the host response generated by the initial SARS-CoV-2 virus infection generated protection from subsequent viral infections for a period of time. Other theories include different modes of transmission or dependence for continued circulation on global air travel for influenza but not for other viruses. There are a number of observational studies and other clinical studies that have explored this question, including a study performed among Singapore military recruits from 2009 to 2014, before the pandemic, which showed that a prior adenovirus (ADV) or influenza virus infection conferred cross-protection against subsequent febrile respiratory infections [15]. Another study, based on England’s SGSS (Second Generation Surveillance System) and the Respiratory DataMart System (RDMS), conducted over 17 weeks, showed that the risk of testing positive for SARS-CoV-2 was 58% lower among influenza-positive cases. However, the small number of patients with a simultaneous coinfection had a risk of death 5.92 times greater than those with neither influenza nor SARS-CoV-2 [16]. 

We performed a systematic review and meta-analysis to determine if the disappearance of influenza was indeed at least partly due to the widespread circulation of the pandemic SARS-CoV-2, which provided some cross-protection against influenza virus infections. The aim of this systematic review was to determine whether prior infection with one respiratory virus provides protection against subsequent different respiratory viral infections. The systematic review is registered in PROSPERO, and the article search process is described below. 

## 2. Materials and Methods

The study protocol was registered with The International Prospective Register of Systematic Reviews (PROSPERO), the major systematic review registry, under the identifier ID: CRD42022295380 [17]. The conduct and reporting of this review adhered to the guidelines outlined in the Preferred Reporting Items for Systematic Reviews and Meta-analyses 2020 (PRISMA 2020) [18].

### 2.1. Search Strategy and Data Sources

For this systematic review, we searched for scientific publications on Pubmed, Embase.com, Scopus, and Cochrane Library from 1960 to 3 September 2022. We included bibliographic databases, reference lists of eligible studies and review articles, key journals, conference proceedings, trial registers, internet resources, and contact with study investigators and experts. The search terms employed for each database are outlined in Table 1 and are designed to encompass all relevant studies. Our search was limited to English-language publications. Initial screening involved assessing titles, abstracts, and methods for relevance against the eligibility criteria. Identified studies were then imported into Rayyan and Endnote for deduplication and further selection. Two independent researchers (V.G. and O.H.C.) conducted the screening process and jointly reviewed each article. Subsequently, these investigators individually evaluated the full texts of records deemed eligible for inclusion. Any discrepancies were resolved through discussion with another co-author (P.A.T.).

### 2.2. Eligibility Criteria

We included both prospective and observational studies (including a retrospective cohort, prospective observational, prospective cohort, observational cohort, analytical, and case–control studies). Case reports and reviews were excluded. Inclusion criteria included patients diagnosed with any of the following respiratory viral infections: influenza, seasonal human coronavirus (HCoV), enterovirus/rhinovirus (EV/RV), ADV, human parainfluenza (HPIV), with laboratory confirmation in the inpatient and outpatient settings. Studies that focused on vaccine evaluations, sero-epidemiology, laboratory techniques used to diagnose respiratory viral infections, non-viral respiratory infections, and non-respiratory viral infections, or those that did not describe cohorts with estimates of the risk of contracting a subsequent infection among those with a previous infection, were excluded. Searches were re-run prior to the final analysis. 

### 2.3. Data Extraction and Management

Relevant data, including study design and methodology, participant demographics, the number of prior respiratory infections (type of virus), and the number of subsequent infections (type of virus), were extracted to an Excel spreadsheet. One person extracted the data (V.G.), and another person (O.H.C.) independently checked the extracted data. Both reviewers resolved disagreements regarding extracted data and achieved consensus by either engaging in discussion with each other or a third reviewer (P.A.T.). Data pertaining to the constructed indices were extracted from all eligible papers, encompassing details such as the first author’s name, publication date, country, study type, age, sex, comorbidities of reinfected patients, the proportion of reinfected patients among discharged individuals, time interval between initial and subsequent clinical symptoms, vaccination status, and patient outcomes.

### 2.4. Outcomes

The primary outcome was the prevalence of any sequential respiratory viral infections amongst those with an initial positive viral swab.

The secondary outcomes included

The prevalence of a different sequential viral infection amongst those with an initial positive swab.The prevalence of specific sequential viral infections is grouped by types of initial viral infection.The pooled estimate of the relative risk of SARS-CoV-2 infection following an initial positive viral swab compared with a negative swab.The pooled prevalence of a sequential viral infection in the subgroup excluding studies on SARS-CoV-2.

### 2.5. Statistical Methods

For the meta-analyses, we used an estimated random effects model, which estimated the pooled prevalence for each specific infectious pathogen following the initial viral infection. The random-effects method was applied in view of the differences in study designs, patient populations, and analytical methods used. Subsequently, the chi-squared tests, I^2^, and Cochran’s Q test were reported to determine heterogeneity. An I^2^ value lower than 25% was interpreted as signifying a low level of heterogeneity. The results were summarized using forest plots, and funnel plots were generated to assess for publication bias.

Analyses were then conducted for the following groups, namely (i) those with an initial negative swab compared with a positive swab, (ii) comparing the prevalence of different types of sequential viral infections amongst those with an initial positive viral swab, and (iii) examining the prevalence of sequential viral infections stratified by types of initial viral infection. For the studies evaluating SARS-CoV-2, we also (iv) estimated the pooled relative risk for SARS-CoV-2 infections following an initial positive swab compared with an initial negative swab. Finally, we also estimated the (v) pooled prevalence of a sequential viral infection in the subgroup, excluding studies on SARS-CoV-2.

All data analyses were performed on SPSS version 20.0 (SPSS, Inc., Chicago, IL, USA) and Microsoft Excel 2019. The analyses were also conducted at a 5% level of significance or its equivalence with 95% confidence intervals. The quality of the studies was assessed using the Newcastle–Ottawa scale [19].

## 3. Results

The initial article search process using precise terms and keywords yielded 7388 records. After removing 6308 duplicates via Endnote and Rayyan software, up to the date of 3 September 2022, 1066 relevant studies on Medline (PubMed), 15 on Embase.com, 2351 on Scopus, 144 on Cochrane Library, and 15 from other sites were identified and obtained. After 1080 articles were screened via the titles and abstracts, and 789 articles were excluded, 291 papers were sorted for retrieval, of which 236 could not be retrieved because of accessibility issues, indexing and abstracting limitations, database coverage, inconsistent citations, and document type exclusions. From these, 55 articles were assessed for eligibility based on the inclusion and exclusion criteria listed above. Finally, six studies were finalized for the meta-analysis (Figure 1). Of the six studies included, a majority were retrospective in nature (n = 5), and two originated from Asia (Singapore and Saudi Arabia), while the remaining studies originated from Europe or the United States. Four studies examined SARS-CoV-2 as the second viral infection following an initial positive or negative viral swab. A summary of these studies is found in Table 2.

### 3.1. Assessment of Quality of Included Studies

The quality of six studies (Gombar, Ringlander, Chen, Khan, Sagar, and Most) was evaluated using the Newcastle–Ottawa Scale (NOS) [20], shown in Table 3. All studies scored highly in several key areas, including representativeness of the exposed cohort, selection of the non-exposed cohort, ascertainment of exposure, demonstration that the outcome of interest was not present at the start of the study, comparability of cohorts based on the design or analysis, and assessment of outcome. These aspects suggest strong methodological rigor and reliability in these studies. However, disparities were observed in the follow-up duration and adequacy of the follow-up of the cohorts, with studies by Gombar, Ringlander, Chen, and Khan, demonstrating sufficient follow-up periods, whereas both the Sagar and Most studies fell short in this aspect. Overall, the Gombar, Ringlander, Chen, and Khan studies received high total scores of eight, indicating a robust study quality, while the Sagar and Most studies scored slightly lower at five and six, respectively, highlighting potential limitations in the follow-up duration and adequacy.

### 3.2. Comparing Subsequent Infections in Patients with an Initial Positive Swab versus a Negative Swab

Amongst the studies with a positive initial viral swab for any virus, the pooled prevalence of a sequential viral infection was 0.08 (95% CI 0.05–0.12) (Figure 2A). There was, however, significant heterogeneity amongst the studies, with a Cochran’s Q = 271.87, *p* < 0.01, and I^2^ = 96%. The corresponding funnel plot also reflected some degree of publication bias (Appendix A). By comparison, when looking at the prevalence of viral infections following an initial negative swab, the pooled prevalence was higher at 0.15 (95% CI 0.00–0.42) (Figure 2B); however, the findings are also limited by heterogeneity amongst the studies (Cochran’s Q = 712.50, *p* < 0.01; I^2^ = 100%) and publication bias (Appendix A).

### 3.3. Is There an Impact of Initial Infection on the Type of Secondary/Sequential Viral Infection?

For the studies that had an initial positive viral swab, we compared the type of sequential viral infection. EV/RV had a pooled prevalence of 0.01 (Figure 3 95% CI 0.00–0.03; Cochran’s Q = 693.58, *p* < 0.001, I^2^ = 98%). By comparison, HCoV (Figure 4; 0.01; 95% CI 0.00–0.03; Cochran’s Q = 693.58, *p* < 0.01; I^2^ = 98%) had a similar pooled prevalence than EV/RV as a sequential infection. However, the findings were similarly limited by significant heterogeneous studies and a probable degree of publication bias (Appendix A).

### 3.4. Does the Type of Initial Viral Infection Matter?

We subsequently examined subgroups of the studies stratified by the type of initial viral infection. Amongst the four studies that examined patients with an initial HCoV viral infection, the pooled prevalence of a subsequent viral infection was 0.09 (Figure 5; 95% CI 0.04–0.17, Cochran’s Q = 43.71, *p* < 0.001, I^2^ = 93%). The corresponding funnel plot is shown (Appendix A). There were two studies with an initial EV/RV infection, and the pooled prevalence of a subsequent viral infection was comparable to that of HCoV, at 0.07 (Figure 6; 95% CI 0.03–0.11, Cochran’s Q = 17.76, *p* < 0.01; I^2^ = 94%). There were also another two studies with an initial influenza infection, indicating a slightly higher pooled prevalence of a sequential viral infection afterward, at 0.17 (Figure 7; 95% CI 0.00–0.66, Cochran’s Q = 32.23, *p* < 0.01, I^2^ = 97%). The findings were similarly limited by a high degree of heterogeneity among the studies. 

### 3.5. What Is the Relative Risk of SARS-CoV-2 Infection Following an Initial Positive Viral Swab Compared with a Negative Swab?

We examined the subgroup of four studies that had SARS-CoV-2 as the outcome of their second viral swab. Among this subgroup, the initial viral swab was positive for either HCoV, MERS-CoV, or RV, or it was negative. The pooled estimate of relative risk for a SARS-CoV-2 infection following an initial positive swab was 0.84 (Figure 8, 95% CI 0.67–1.07, Cochran’s Q 4.07, *p* = 0.25, I^2^ = 26%). These findings were limited by a moderate degree of heterogeneity among the studies and publication bias (Funnel Plot shown as Appendix A).

### 3.6. Subgroup Analysis Excluding Studies on SARS-CoV-2, Showing Pooled Prevalence of Viral Infections Following an Initial Positive Swab

Excluding the studies that examined SARS-CoV-2, the pooled prevalence of a secondary viral infection following an initial positive viral swab was 0.06 (Figure 9, 95% CI 0.03–0.09, Cochran Q 74.66, *p* < 0.01, I^2^ = 91%). The corresponding funnel plot shows a degree of publication bias (Appendix A).

## 4. Discussion

In this systematic review and meta-analysis, we hypothesized that a prior respiratory viral infection may protect an individual from a subsequent different respiratory viral infection. This could be due to a broad host immune response generated by the initial infection, which may protect the person from a subsequent infection for a period of time. A few other studies have also suggested that infections with certain viruses provide relative cross-protection to other unrelated viruses, though not to all of them [21,22,23,24,25,26]. 

Overall, our review found an 8% probability of subsequent confirmed respiratory viral infection after an initial positive viral test, contrasted with a 15% chance following a negative initial swab test. This supports our original hypothesis that a broad immune response from an initial respiratory viral infection may protect from another infection for up to3 years, depending on the primary and secondary virus. However, it is important to note that the reliability of this result may be limited as all three studies for the negative initial swab test solely examined SARS-CoV-2 and not any of the other seasonal viruses. 

In terms of secondary infections, the most common secondary infection was from EV/RV, with a prevalence of 0.01. However, as mentioned, most studies focused on SARS-CoV-2 as the second viral swab, with the initial viral swab results encompassing either a positive result for HCoV, MERS-CoV, or RV, or a negative result for any virus. The pooled estimate of relative risk for acquiring a SARS-CoV-2 infection subsequent to a positive initial respiratory viral swab was 0.84, which may reflect the extremely high attack rates of SARS-CoV-2 worldwide. These findings were also constrained by a moderate level of heterogeneity across the studies and evidence of publication bias. When excluding studies examining SARS-CoV-2 specifically, the combined prevalence of secondary respiratory viral infections following an initial positive viral swab was calculated at 0.06.

Some studies that did not meet our inclusion criteria (i.e., they did not compare individual patients) describe some interesting findings about other coronaviruses. For example, the severity of COVID-19, as indicated by hospitalization and/or referral to intensive care, was not significantly decreased by previously confirmed seasonal HCoV infections in two studies [27,28]. However, the survival probability was significantly higher among hospitalized COVID-19-positive patients in the HCoV-positive group compared to those in the HCoV-negative group [29].

The opposite was found when looking at the effect of previous MERS positivity on the severity of acute COVID-19 infections. In a Saudi study among those who contracted COVID-19, individuals in the MERS-positive group faced a significantly higher risk of hospitalization and mortality in comparison to the MERS-negative group, although this may reflect comorbidities. Indeed, after adjusting for age and sex, the risk of mortality was found to be similar between the two groups [30]. 

Multiple animal and epidemiological modeling studies have shown that a single respiratory infection can influence subsequent immune responses to unrelated pathogens at the same site, with effects that persist over an extended period [31,32]. A study of “older” centenarians exposed to the “Spanish flu” of 1918 showed significantly lower COVID-19 mortality rates compared to “younger” centenarians. The authors speculated that the enduring presence of cross-reactive immune mechanisms throughout their lifetimes may have empowered centenarians previously exposed to the Spanish flu to effectively combat the threat of COVID-19 a century later [33].

The phenomenon known as “innate imprinting” or “innate education” is one possible mechanism to reduce the risk of illness associated with respiratory viral infections or to diminish inflammation and mitigate immunopathology. This may underlie our finding of reduced second viral infections, but more mechanistic studies with in-depth laboratory studies need to be undertaken to prove this hypothesis.

One strength of this study is that we included the most common major viruses (influenza, seasonal coronaviruses, enterovirus/rhinovirus, adenovirus, respiratory syncytial virus, parainfluenza, and SARS-CoV-2) from different studies to look for the association between a prior infection, which may protect patients from a subsequent infection (either reinfection and/ or different viral infection). 

However, there are significant limitations, especially the heterogeneity within the included studies; a low number of studies have looked at documented sequential infections in individual patients rather than broad sero-epidemiology studies, and there is the potential for publication bias. For example, the probabilities of subsequent infection after initial infections with HCoV and EV/RV were found to be lower at 0.09 and 0.07, respectively, compared to influenza, which had a 0.17 probability of subsequent infections in patients. This may have been influenced by the higher availability of the influenza virus and SARS-CoV-2 testing and low pre-2019 immunity to SARS-CoV-2 compared with other viruses.

### Limitations

Firstly, the NOS score indicated a high likelihood of bias in the studies selected, given the relatively small number of studies and risk for publication bias in this context (Table 3). As we wanted to examine the impact of laboratory-confirmed infections, we only included studies that reported sequential viral swabs. The practice of obtaining viral swabs for testing depends on the clinical practice, leading to a potential bias towards testing for SARS-CoV-2 or influenza. Secondly, the time interval between the first and second infections was not clearly stated in some of the included studies. Furthermore, it is important to acknowledge that the prevalence of SARS-CoV-2 was likely influenced by various factors, including it being a new virus for which everyone was initially naïve, different waves, etc., potentially impacting the outcomes of studies utilizing negative first-swab data. While recognizing the heterogeneity in the timing of the second swab and the dynamic nature of the pandemic, it is also important to understand that studies exclusively focusing on COVID-19 may not fully capture the broader spectrum of other respiratory viral infections. Our detection of a small potential signal echoing the findings of Chen et al. [15] is still notable. 

## 5. Conclusions

This systematic review and meta-analysis provided a preliminary basis for exploring the interactions among various respiratory viruses, possibly indicating a protective effect against subsequent infections in individuals with prior respiratory viral infections compared to those with negative respiratory viral swabs. Understanding these interactions, including the role of a broad innate immune response, is crucial not only for enhancing preparedness for future pandemics but also for preventing seasonal respiratory infections via vaccines and immunotherapies.

## Figures and Tables

**Figure 1 viruses-16-00982-f001:**
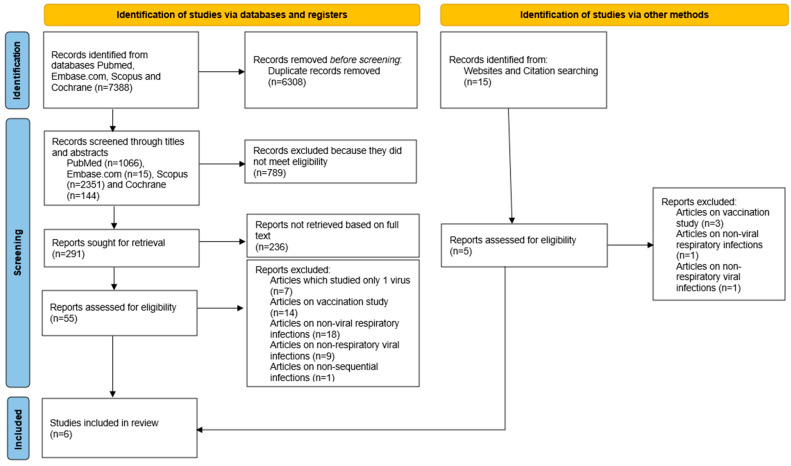
PRISMA 2020 flow diagram for systematic reviews [18].

**Figure 2 viruses-16-00982-f002:**
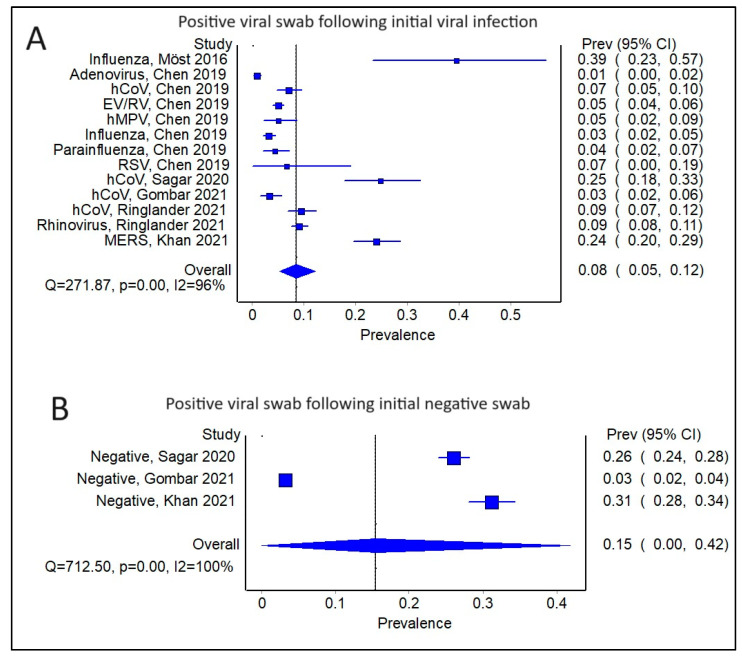
(**A**) Forest plot showing pooled prevalence of a positive viral swab (any virus) following an initial viral infection (any virus) (**B**) Forest plot showing pooled prevalence of a positive viral swab (any virus) following an initial negative swab.

**Figure 3 viruses-16-00982-f003:**
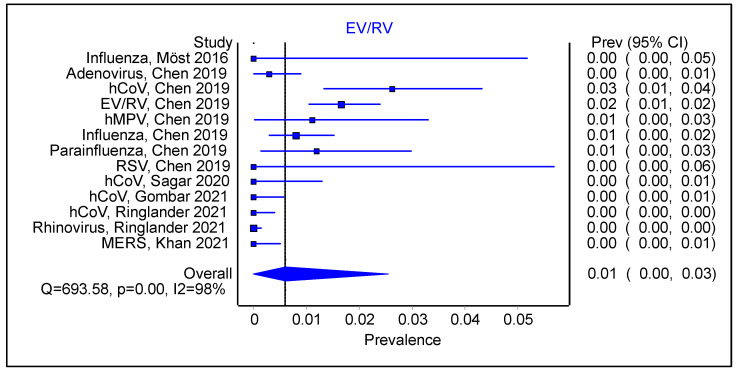
Forest plot showing pooled prevalence of enterovirus/rhinovirus infection, following an initial positive viral swab.

**Figure 4 viruses-16-00982-f004:**
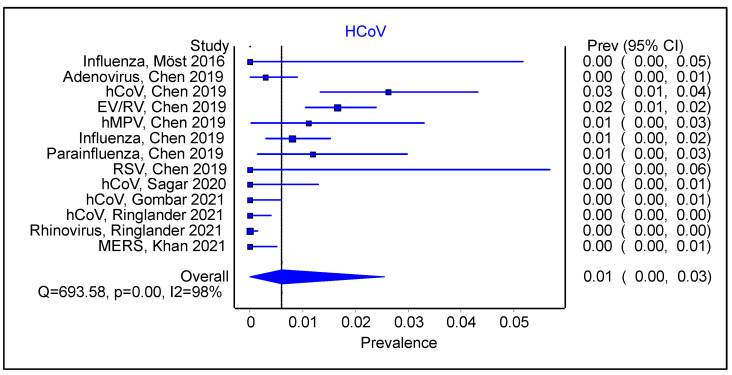
Forest plot showing pooled prevalence of seasonal coronavirus infection following an initial positive viral swab.

**Figure 5 viruses-16-00982-f005:**
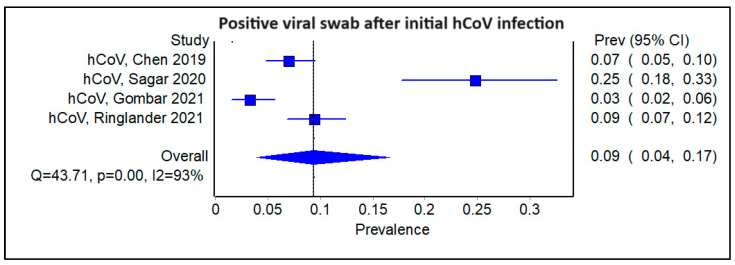
Forest plot showing pooled prevalence of a positive viral swab following an initial seasonal coronavirus infection.

**Figure 6 viruses-16-00982-f006:**
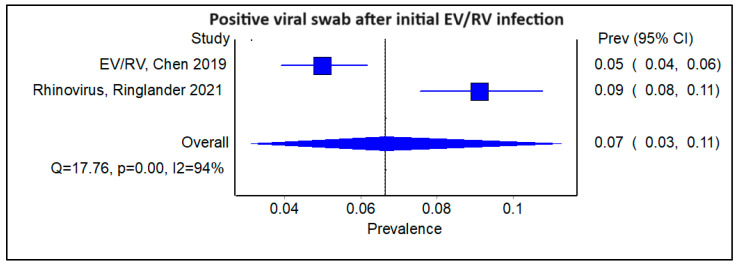
Forest plot showing pooled prevalence of a positive viral swab following an initial enterovirus/rhinovirus infection.

**Figure 7 viruses-16-00982-f007:**
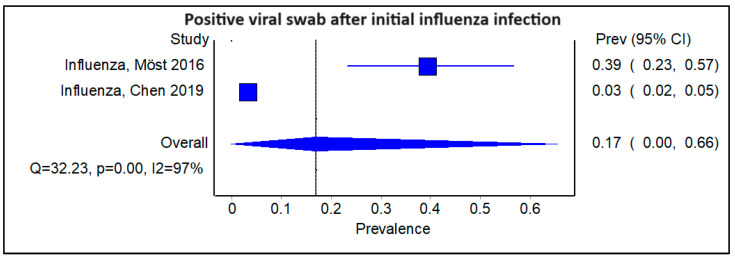
Forest plot showing pooled prevalence of a positive viral swab following an initial influenza infection.

**Figure 8 viruses-16-00982-f008:**
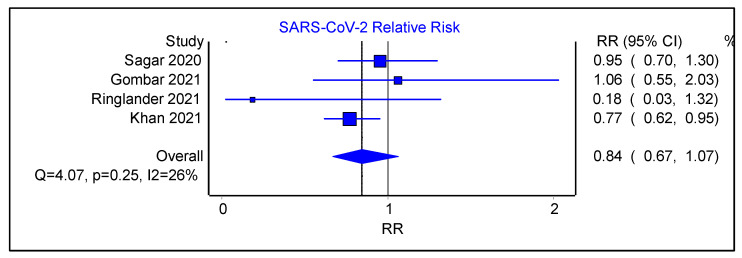
Forest plot showing pooled estimate of relative risk of SARS-CoV-2 following an initial positive viral swab.

**Figure 9 viruses-16-00982-f009:**
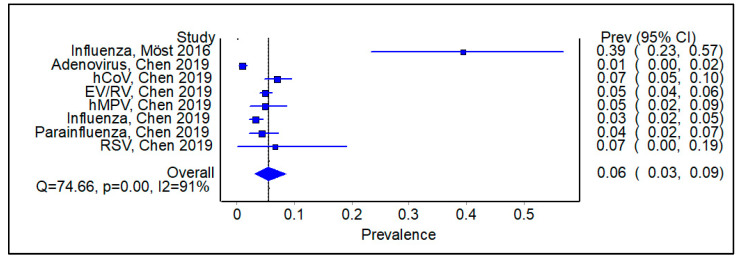
Subgroup analysis of studies excluding SARS-CoV-2, showing pooled prevalence of a secondary viral infection following an initial positive viral swab.

**Table 1 viruses-16-00982-t001:** Search strategies as run in each database adhering to PRISMA-S checklist.

Database: Medline (PubMed) Date of Search: 13 August 2022 No. of Results: 1066 Limits and filters applied (referring to PRISMA-S item 9 & 10): nil
Search Strategy (copy and paste as run) **(((respiratory tract infections[MeSH Terms]) OR (“Respiratory tract infection*”[Title/Abstract] OR “Respiratory viral infection*”[Title/Abstract] OR “Viral respiratory infection*”[Title/Abstract] OR Influenza[Title/Abstract] OR Flu[Title/Abstract] OR H1N1[Title/Abstract] OR H3N2[Title/Abstract] OR “Respiratory Syncytial virus*”[Title/Abstract] OR RSV[Title/Abstract] OR Adenovirus[Title/Abstract] OR Parainfluenza[Title/Abstract] OR Enterovirus[Title/Abstract] OR Rhinovirus[Title/Abstract] OR Coronavirus*[Title/Abstract] OR COVID-19[Title/Abstract] OR SARS CoV-2[Title/Abstract])) AND ((cross protection[MeSH Terms]) OR (“Cross-protect*”[Title/Abstract] OR “cross protect*”[Title/Abstract] OR co-infect*[Title/Abstract] OR “virus cross-protect*”[Title/Abstract] OR “virus cross protect*”[Title/Abstract] OR “cross-reacti*”[Title/Abstract]))) AND ((((((“observational stud*”[Publication Type]) OR (observation[MeSH Terms])) OR (cohort studies[MeSH Terms])) OR (cohort analyses[MeSH Terms])) OR (case control studies[MeSH Terms])) OR (case control studies[MeSH Terms]))**
Database: Embase.com Date of Search: 13 August 2022 No. of Results: 15 Limits and filters applied (referring to PRISMA-S item 9 & 10): nil
Search Strategy (copy and paste as run) (‘viral respiratory tract infection’/exp OR (‘respiratory tract infection’:ab,ti OR ‘respiratory tract viral infection’:ab,ti OR ‘respiratory tract virus infection’:ab,ti OR ‘respiratory viral infection’:ab,ti OR ‘viral respiratory disease’:ab,ti OR ‘viral respiratory tract infection’:ab,ti)) AND (‘heterologous immunity’/exp OR (‘cross immunity’:ab,ti OR ‘cross protection immunity’:ab,ti OR ‘cross serotype immunity’:ab,ti OR ‘cross serovar immunity’:ab,ti OR ‘cross variant immunity’:ab,ti OR ‘cross-protective immunity’:ab,ti OR ‘cross-reactive immunity’:ab,ti OR ‘cross-strain immunity’:ab,ti OR ‘crossprotective immunity;’:ab,ti OR ‘crossreactive immunity’:ab,ti OR ‘hetero-subtype immunity’:ab,ti OR ‘hetero-subtypic immunity’:ab,ti OR ‘heterogeneous immunity’:ab,ti OR ‘heterogenous immunity’:ab,ti OR ‘heterospecific immunity’:ab,ti OR ‘heterosubtype immunity’:ab,ti OR ‘heterosubtypic immunity’:ab,ti OR ‘heterovariant immunity’:ab,ti OR ‘immunity, heterologous’:ab,ti))
Database: Scopus Date of Search: 13 August 2022 No. of Results: 2351 Limits and filters applied (referring to PRISMA-S item 9 & 10): nil
Search Strategy (copy and paste as run) (TITLE-ABS-KEY(Cross-protection OR “cross protection” OR “heterologous immunity” OR “Cross-protect*” OR “cross protect*” OR co-infect* OR “virus cross-protect*” OR “virus cross protect*” OR “cross-reacti*”)) AND (TITLE-ABS-KEY(“Respiratory tract infection*” OR “Respiratory viral infection*” OR “Viral respiratory tract infection” OR “Viral respiratory infection*” OR Influenza OR Flu OR H1N1 OR H3N2 OR “Respiratory Syncytial virus*” OR RSV OR Adenovirus OR Parainfluenza OR Enterovirus OR Rhinovirus OR Coronavirus* OR COVID-19 OR SARS-CoV-2))
Database: Cochrane Library Date of Search: 13 August 2022 No. of Results: 144 Limits and filters applied (referring to PRISMA-S item 9 & 10): nil
Search Strategy (copy and paste as run) #1 MeSH descriptor: [Respiratory Tract Infections] explode all trees 18,339 #2 (“Respiratory tract infection*” OR “Respiratory viral infection*” OR “Viral respiratory tract infection” OR “Viral respiratory infection*” OR Influenza OR Flu OR H1N1 OR H3N2 OR “Respiratory Syncytial virus*” OR RSV OR Adenovirus OR Parainfluenza OR Enterovirus OR Rhinovirus OR Coronavirus* OR COVID-19 OR SARS-CoV-2):ti,ab,kw 30,577 #3 MeSH descriptor: [Cross Protection] explode all trees 20 #4 (Cross-protection OR “cross protection” OR “heterologous immunity” OR “Cross-protect*” OR “cross protect*” OR co-infect* OR “virus cross-protect*” OR “virus cross protect*” OR “cross-reacti*”):ti,ab,kw 1125 #5 (#1 OR #2) AND (#3 OR #4) 144

**Table 2 viruses-16-00982-t002:** Summary of included studies (n = 6).

First Author	Year	Country	Study Type	Total Samples	1° Pathogen	2° Pathogen	P+/S+ *	Interval Between 1° & 2° Pathogen	P+/ S− *	P−/ S+ *	P−/ S− *
Gombar	2021	USA	Retrospective	2768	HCoV	SARS-CoV-2	10	NS	292	77	2389
Ringlander	2021	Sweden	Retrospective	434	HCoV	SARS-CoV-2	41	NS	393	0	0
				1242	RV	SARS-CoV-2	113	NS	1129	0	0
Chen	2019	Singapore	Prospective	677	ADV	ADV	2	Days: Median 34 (IQR 21–55)	675	0	0
				677	ADV	HCoV	1		676	0	0
				677	ADV	EV/RV	3		674	0	0
				458	HCoV	ADV	5		453	0	0
				458	HCoV	HCoV	7		451	0	0
				458	HCoV	EV/RV	11		447	0	0
				458	HCoV	HMV	3		455	0	0
				458	HCoV	Flu	2		456	0	0
				458	HCoV	HPIV	4		454	0	0
				1449	EV/RV	ADV	15		1434	0	0
				1449	EV/RV	HCoV	16		1433	0	0
				1449	EV/RV	EV/RV	24		1425	0	0
				1449	EV/RV	HMV	3		1446	0	0
				1449	EV/RV	Flu	8		1441	0	0
				1449	EV/RV	HPIV	4		1445	0	0
				1449	EV/RV	RSV	2		1447	0	0
				181	HMV	ADV	1		180	0	0
				181	HMV	HCoV	2		179	0	0
				181	HMV	EV/RV	4		177	0	0
				181	HMV	Flu	1		180	0	0
				181	HMV	HPIV	1		180	0	0
				870	Flu	ADV	4		866	0	0
				870	Flu	HCoV	8		862	0	0
				870	Flu	EV/RV	8		862	0	0
				870	Flu	Flu	5		865	0	0
				870	Flu	HPIV	3		867	0	0
				254	HPIV	ADV	1		253	0	0
				254	HPIV	HCoV	2		252	0	0
				254	HPIV	EV/RV	3		251	0	0
				254	HPIV	Flu	3		251	0	0
				254	HPIV	HPIV	2		252	0	0
				30	RSV	EV/RV	1		29	0	0
				30	RSV	Flu	1		29	0	0
Khan	2021	Saudi Arabia	Retrospective	1176	MERS-CoV	SARS-CoV-2	82	Years: Median 3.4 (IQR 3.6)	260	260	574
Sagar	2020	USA	Retrospective	1812	HCoV	SARS-CoV-2	33	Days: Median 121 (IQR 69–440)	100	437	1242
Möst	2016	Austria	Retrospective	33	Flu A	Flu B	13	Days: Mean 50	20	0	0

* Result of test for first viral infection (primary/P) and sequential viral infection (secondary/S): P+/S+, Test for both primary and secondary pathogen positive; P+/S−, Test for primary pathogen positive but test for secondary pathogen negative; P−/S+, Test for primary pathogen negative but test for secondary pathogen positive; P−/S−, Test for both primary and secondary pathogen negative. Abbreviations: NS, Not Stated; IQR, Interquartile range; HCoV, Seasonal Human Coronavirus; RV, Rhinovirus; EV/RV, Enterovirus/Rhinovirus; HPIV, Human Parainfluenza Virus; HMV, Human Metapneumovirus; RSV, Respiratory Syncytial Virus; ADV, Adenovirus; Flu, Influenza.

**Table 3 viruses-16-00982-t003:** Assessment of the quality of the included studies—Newcastle–Ottawa Scale.

NOS Items	Gombar	Ringlander	Chen	Khan	Sagar	Most
Representativeness of the exposed cohort	1	1	1	1	0	1
Selection of the non-exposed cohort	1	1	1	1	1	1
Ascertainment of exposure	1	1	1	1	1	1
Demonstration that outcome of interest was not present at the start of the study	1	1	1	1	1	1
Comparability of cohorts on the basis of the design or analysis	1	1	1	1	1	1
Assessment of outcome	1	1	1	1	1	1
Follow-up was long enough for outcomes to occur	1	1	1	1	0	0
Adequate of follow-up of cohorts	1	1	1	1	0	0
**Total Score**	8	8	8	8	5	6

## Data Availability

The original contributions presented in the study are included in the article/Appendix A, further inquiries can be directed to the corresponding authors.

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
