# Peer review of "Does Prior Respiratory Viral Infection Provide Cross-Protection Against Subsequent Respiratory Viral Infections? A Systematic Review and Meta-Analysis"

_viruses, 2024, doi:10.3390/v16060982_

Round 1

Reviewer 1 Report

Comments and Suggestions for Authors

The problem of co-infections and subsequent infections is global, and nowadays is especially exciting with respect to the SARS-CoV-2 pandemics. Thus, importance of pre-existing immune responses to seasonal endemic coronaviruses (HCoVs) for the susceptibility to SARS-CoV-2 infection and the course of COVID-19 is the subject of an ongoing scientific debate. 

The authors made an attempt to summarize and analyse the data regarding impact of the previous respiratory viral infections into the susceptibility and development of the subsequent respiratory viral infections.

The authors seem to have done a great job of data collection and meta-analysis, but the result is (in my opinion) disappointing. Actually, there is no answer provided to the question the authors have asked themselves in the Introduction:

"The aim of this systematic review was to determine whether prior infection with one respiratory virus provides protection against subsequent respiratory viral infections."

Instead we read in the Abstract that: "However, significant heterogeneity and publication bias were noted."

Thus, this review can be considered more as a "methodological" work and should probably be published, although after the major revision.

My main complaints are as follows:

1. I did not quite understand why the authors have collected only 6 papers from for their meta-analysis from the "1066 relevant studies on Medline (PubMed), 15 on Embase.com, 2351 on Scopus, 144 on Cochrane Library and 7 from other sites were identified obtained until 3rd of September 2022 (1080 articles in total)?

2. Did the authors have calculated "significant heterogeneity and publication bias" among those 6 papers?

3. Is there a list of those initially relevant 1080 papers available?

E.g. I know of one good article that is quite relevant to this topic, which was not referenced in the list of references.

doi: 10.1016/j.celrep.2021.110169

By what criterion did the authors ignore it? How can I be sure that other important papers have not been ignored as well?

4. Please add in the conclusion some suggestions that could improve the analysis of the problem of "subsequent infections" in the future work in order to get more positive / unambiguous results. What new design/criteria/etc. should be added for meta-analysis?

There are also some minor issues:

Please check if all abbreviations are deciphered.

E.g. I did not find the deciphering for some viruses: EV; HCoV; ADV; HPIV as well as for PROSPERO (or provide a website for the resource).

Reviewer 2 Report

Comments and Suggestions for Authors

This manuscript presents the results of a systematic review and meta-analysis of published papers on the prevalence of a secondary respiratory viral infection after contracting a primary respiratory virus. The authors did great job analyzing all published literature on this topic and selected six original studies to draw some conclusions on the possibility of a secondary viral infection. This is not surprising that considerable variation in the data was found and clear conclusions are difficult to draw.

A major weakness of the study is that it does not discuss the timing of when reinfection occurred. It is known that the non-specific innate responses during first days of the initial infection will protect against reinfection with other viruses. Were the timeframes between primary and secondary infections discussed in the original studies?

Minor comments:

1.       The first paragraph in the Introduction section doesn’t have any relevant reference.

2.       Please give the definitions of the abbreviations of virus names wherever they are mentioned for the first time

3.       Lanes 40-45: please re-phrase the sentence, as the words “nonpharmaceutical interventions” are given twice here.

4.       Lane 70. Please re-phrase the sentence, as it is not clear what “if this is the case” refers to.

5.       Table 1 shows the results of the study, not Methods. It should be moved to the relevant section.

6.       there are other grammatical and punctuation errors in the text, a thorough check of the text by an English-speaking editor is required

7.       lanes 179-181: please provide as reference number, not the full citation

8.       Figures 2, S1: the quality of the figures needs improvement

9.       Lanes 273-275: this sentence has repeats the word “some” three times, please correct

Comments on the Quality of English Language

hThere are multiple grammatical and punctuation errors in the text, a thorough check of the text by an English-speaking editor is required

Round 2

Reviewer 1 Report

Comments and Suggestions for Authors

The review was approved. Most of my questions have been answered. There are no further comments.

Reviewer 2 Report

Comments and Suggestions for Authors

The authors substantially revised their manuscript and adequately addressed all issues raised during original peer review.